# Implementation and impact of mhealth in the management of diabetes mellitus in Africa: A systematic review and meta-analysis

**Franklin Okechukwu Dike**[1]*, **Jean Claude Mutabazi**[2], **Ezekiel Musa**[3,4], **Blessing Chinenye Ubani**[1,5], **Ahmed Sherif Isa**[6,7], **Chidiebele Malachy Ezeude**[8,9], **Henry Iheonye**[10], **Isah Idris Ainavi**[11,12]

**1** University of Uyo Teaching Hospital, Internal Medicine, Akwa Ibom, Nigeria, **2** Department of Social and Preventive Medicine, École de Santé Publique, Université de Montréal, Canada, **3** Kaduna State University, Department of Internal Medicine, Kaduna, Nigeria, **4** University of Cape Town, Department of Medicine, Cape Town, South Africa, **5** University of Uyo, Internal Medicine, Akwa Ibom, Nigeria, **6** Stellenbosch University, Faculty of Medicine and Health Sciences, Department of Anesthesiology and Critical Care, Stellenbosch, South Africa, **7** Ahmadu Bello University, Human Physiology, Zaria, Nigeria, **8** Nnamdi Azikiwe University, Internal Medicine, Awka, Anambra, Nigeria, **9** Nnamdi Azikiwe University Teaching Hospital, Internal Medicine, Awka, Anambra, Nigeria, **10** Federal Teaching Hospital Lokoja, Internal Medicine, Kogi, Nigeria, **11** Kaduna State University, Chemical Pathology and Immunology, Kaduna, Nigeria, **12** Barau Dikko Teaching Hospital Kaduna, Chemical Pathology, Kaduna, Nigeria

* frankincense4m@uuthuyo.net, frankincense4m@gmail.com

## Abstract

### Background

The World Health Organization (WHO) has proposed the concept of mobile health to support healthcare systems delivery worldwide. Mobile health (mHealth) involves using Information and Communication Technology (ICT) for health care provision or delivery services. In the context of Africa, a region that has witnessed a significant increase in mobile phone availability and usage in the last decade and a corresponding rise in the incidence and prevalence of diabetes mellitus, this study has global implications. We conducted a systematic review on the extent of mHealth implementation in managing diabetes mellitus in Africa. We estimated its impact on achieving desired glycemic targets, sustained control, and preventing complications in the past decade.

### Methods and analysis

The studies assessing the utilization of mHealth in managing patients with diabetes mellitus in Africa were considered based on the PICO method: Population, Intervention, Comparator, and Outcomes. MEDLINE, PubMed, SCOPUS, and the Pan African Clinical Trials Registry were searched. Two authors, independent of each other, screened titles and abstracts retrieved using the search strategy, retrieved the full-text articles, and assessed them for eligibility, extracting data after that. A third independent reviewer was brought in to resolve disagreements between the two authors by discussion. The revised Cochrane Collaboration Risk of Bias Tool was used to assess the quality of included studies. A narrative synthesis of extracted data was done due to the paucity of eligible studies, and the results were summarized in a meta-analysis.

**Data availability statement:** All data is available without restrictions at Open Science framework (OSF) and can be accessed through the following links; results (https://osf.io/ahs5k/?view_only=5e8f337abbff-45bab7f45789a23a74a3 OR https://osf.io/ahs5k/), data (https://osf.io/ets2w/?view_only=f689e62f33794186a65bcd9e1d733a70 OR https://osf.io/ets2w/), analysis (https://osf.io/hg3dy/?view_only=2cf5f9c706a64f-48b6750933aed1e11e OR https://osf.io/hg3dy/), and database search outcomes (https://osf.io/d7xmt/?view_only=9a093f27345d4256b8a3626e0556a2f5 OR https://osf.io/d7xmt/)

**Funding:** The author(s) received no specific funding for this work.

**Competing interests:** The authors have declared that no competing interests exist.

## Results

None of the six included studies measured the mean FPG or percentage changes as primary outcomes. Five measured the percentage change in HbA1c from baseline to the end of the study. The percentage change in HbA1c from the baseline ranged from 3.6% to 20.53%, achieving significance in three studies. In the meta-analysis the overall WMD (95% CI) was 0.992 (0.48, 1.50). This, in combination with a high z score of 3.822, p <0.001 suggests a statistically significant overall effect that is not likely due to chance. However, a considerable heterogeneity ($I^2$ = 63.9%, p = 0.026) was present implying that the observed effect may not be generalizable to all the studies due to differences in study characteristics in this case most likely sample size and duration of study. None of the studies addressed the secondary outcomes of measuring the direct relationships between these mHealth interventions and the prevention or early detection of diabetes complications.

## Conclusion

Overall, there was a statistically significant reduction in HbA1c levels among individuals living with type 2 diabetes in Africa following mHealth interventions. Few studies were included in the meta-analysis with significant heterogeneity. Therefore, we recommend more well-designed randomized controlled trials to investigate the implementation and efficacy of mHealth in the management of diabetes mellitus in Africa.

## Systematic review registration

**PROSPERO** CRD42021218674

## Author summary

In 2009, the World Health Organization (WHO) proposed the concept of mobile health as a strategy to boost the delivery of health care worldwide. Since then, one disease that has received significant adoption and impact from this strategy is diabetes mellitus, and various tools such as text messages, phone calls, mobile applications, and wearable devices have been employed to implement this strategy. Given the current trend of increasing mobile phone availability, uptake, and usage across Africa, we investigated the utilization of this strategy in Africa with an emphasis on managing diabetes mellitus using a systematic review and meta-analysis. We searched relevant databases for randomized controlled studies examining the use of mHealth for patients with diabetes mellitus in Africa. The search yielded nothing for Type 1 diabetes mellitus, but 1481 studies for type 2 diabetes mellitus. Out of these 1481 studies, only six met the inclusion criteria. We conducted a meta-analysis of the changes in glycated hemoglobin from the included studies and found a statistically significant positive effect. However, moderate heterogeneity was observed, indicating these results may not be generalizable or should be interpreted with caution. The key takeaway from our findings is the necessity for further efforts to collect sufficient data, which will enable more comprehensive research and lead to conclusive outcomes.

## Introduction

The Global Observatory for eHealth's 3rd edition in 2011 showed that eHealth was still only emerging in many member states through experimentation in their health sectors. Excellent strategic implementation was observed in high-income countries, with up to 87% implementation of various mHealth categories, compared with low-income and middle-income countries (LMIC), with about 77%. [1]

Mobile health or mHealth was first defined by Robert Istepanian in his treatise in 2006 as "emerging mobile communication and network technologies for healthcare" [2]. Simply put, it describes using mobile and wireless communication technologies to improve healthcare delivery, outcomes and research [3]. In the early days of mHealth, it was mainly a function of mobile communication using SMS or text messaging, phone calls or data exchange over cellular networks, with communication exchange between healthcare providers and consumers as the basis for implementation [4]. However, with the emergence of smartphones following the introduction of the Apple iPhone in 2007, mHealth history has witnessed a paradigm shift into an app/post-app era in which mHealth apps are becoming the mainstay [4]. These mHealth apps have become more valuable following the advances in smartphone operating systems, wearables and sensors [5]. The same feedback is seen both with provider-facing apps and consumer-facing apps. The cornerstones of the post-app era are the advancement in wireless and Bluetooth connectivity, GPS (global positioning systems), improved internet speeds, and the use of sensors and wearables [5]. A Pew Research Center report in 2016 documented that globally, 88% of adults owned a phone, of which 43% were cellular phones and 45% were smartphones.[4] 77% of American adults owned a smartphone compared to 4% of adults in Ethiopia and Uganda [4].

In the past decade, there has been a progressive increase in the penetration and usage of mobile phones in Africa. A recent survey conducted across six countries in Africa including South Africa, Nigeria, Tanzania, Ghana, Senegal, and Kenya, revealed mobile phone penetration rates of 91% in South Africa, 80% in Nigeria, and 75% in Tanzania, the latter being the lowest among the surveyed nations [6]. Intriguingly, there has been a dramatic increase in the prevalence of non-communicable diseases (NCDs) in Africa [7].

Diabetes mellitus, being one of the leading NCDs, remains a critical economic and developmental challenge in Africa, with estimated costs of USD 3.4 billion in 2015 [7]. Therefore, preventing diabetes mellitus and improving self-management among people with diabetes mellitus constitute major public health priorities. mHealth has been thought to be a suitable supportive intervention in managing diabetes mellitus, helping patients achieve desired glycemic targets, sustain control, and prevent complications [8,9].

A 2011 meta-analysis involving 1657 individuals with either type 1 or type 2 diabetes mellitus using SMS messages to send self-monitored blood glucose values and receive self-management information showed a 0.5% reduction in glycated hemoglobin (HbA1c) over six months in mHealth intervention groups compared to the control groups [10]. More recently, another meta-analysis, conducted in 2017 and involving 13 studies, reported a mean reduction of 0.44% in HbA1c in mHealth intervention groups compared to controls [9]. Furthermore, it reported an increased perception of self-care. In 2021, Eberle et al. reported that diabetes-specific mHealth mobile applications significantly improved glycemic control in individuals with type 1 and type 2 diabetes mellitus [11]. The WHO recommends an HbA1c glycemic target of ≤7% for most patients with type 2 diabetes mellitus [12]. Thus, the growing increase in mobile phone usage across Africa is an encouraging trend regarding its potential to support existing overburdened healthcare systems combating both communicable and non-communicable diseases.

No data shows the degree of mHealth implementation in Africa as WHO-GOe recommended, especially regarding managing diabetes mellitus. A systematic review of economic evaluation of mHealth solutions from studies conducted mostly in upper and upper-middle-income countries concluded that mHealth is cost-effective [13]. However, the same cannot be said for the African continent due to the unavailability of data.

We hope this systematic review's outcomes will provide data to guide clinicians, diabetes nurse educators, and dieticians on the benefits and compelling need for implementing mHealth. Furthermore, the data obtained will help develop policy frameworks that aim to incorporate mHealth into the standard of diabetes care in Africa.

## Objectives

The aim of the study to assess the impact of mHealth interventions on managing diabetes mellitus in Africa, including their effectiveness in achieving treatment goals and their influence on early detection and management of diabetic complications. It also sought to identify barriers to mHealth utilization in Africa and evaluate the potential harms and costs of implementing mHealth strategies.

## Methods

**Eligibility criteria.** We considered all published randomized controlled trials and non-randomized controlled trials, quasi-randomized controlled trials and observational studies deploying one or more mHealth strategies in the management of diabetes mellitus in Africa done in the continent between January 2010 and October 2020, including unpublished studies from the Pan African Clinical Trials Registry (PACTR) [14].

All the articles had similar standard cut-off points for diagnosing diabetes mellitus and treatment targets (fasting plasma glucose (FPG) and HbA1c) based on WHO and the International Diabetes Federation (IDF) guidelines [12,15,16].There were no language restrictions; however, all the search articles were in English. We excluded studies testing the acceptability of mHealth among health professionals and duplicate publications.

### Participants

The review considered studies conducted in Africa among diabetes mellitus patients with no restrictions on age or sex. It included all studies reporting diabetes mellitus (including type 1 and type 2 diabetes mellitus).

### Search strategy and selection criteria

We searched relevant electronic databases to identify relevant studies on the implementation of mHealth for Diabetes Mellitus (DM) in Africa, covering the period from January 2010 to October 2020. Specifically, we searched the Medline database (PubMed), the Cochrane Register of Controlled Trials (CENTRAL), Embase (Scopus), and the CINAHL database. Additionally, we explored the Pan African Clinical Trials Registry (PACTR) for ongoing and recently completed but unpublished studies meeting our criteria.

In our search strategy for the Medline database, we utilized Medical Subject Headings (MeSH) to identify all available studies related to mHealth and diabetes mellitus. We included terms representing various mHealth categories and applied filters for diabetes mellitus in African regions and countries.

(implement* OR execute* OR administrate* OR organise* OR fulfill* OR perform* OR utilise* OR manage* OR operate* OR act* OR realise* OR effect* OR impact*) AND (mHealth* OR "mobile health*" OR eHealth* OR "electronic health*") AND (diabetes* OR diabetes

mellitus, type 1/ OR diabetes mellitus, type 2 OR diabetes, gestational/) AND Africa/ OR "Africa South of the Sahara"/ OR "Sub-Saharan Africa"/ OR north Africa/ OR Africa, Northern/ Egypt or Libya OR Tunisia OR Algeria OR Morocco OR "Western Sahara" OR Angola/ OR Benin/ OR Botswana/ OR Burkina Faso/ OR Burundi/ OR Cameroon/ OR Cape Verde/ OR Central African Republic/ OR Chad/ OR Comoros/ OR Congo/ OR Brazzaville/ OR Cote d'Ivoire/ OR Djibouti/ OR Equatorial Guinea/ OR Eritrea/ OR Ethiopia/ OR Gabon/ OR Gambia/ OR Ghana/ OR Guinea/ OR Bissau/ OR Kenya/ OR Lesotho/ OR Liberia/ OR Madagascar/ OR Malawi/ OR Mali/ OR Mauritania/ OR Mauritius/ OR Mozambique/ OR Namibia/ OR Niger/ OR Nigeria/ OR Rwanda/ OR Sao Tome e Principe/ OR Senegal/ OR Seychelles/ OR Sierra Leone/ OR Somalia/ OR South Africa/ OR South Sudan/ OR Sudan/ OR Swaziland/ OR Eswatini OR Tanzania/ OR Togo/ OR Uganda/ OR Western Sahara/ OR Zaire/ OR Zambia/ OR Zimbabwe/.

All available controlled trials using this search were retrieved (S1 Table). Similar methods were used to search through CENTRAL and Embase (Scopus). When necessary, hand searching and snowballing were used through all the databases to retrieve all relevant articles. Data from indexed dissertations were retrieved where possible. The PACTR was also searched for ongoing, recently published or unpublished trials meeting the inclusion criteria. All searches were carried out using the Preferred Reporting Items for Systematic Reviews and Meta-Analyses guidance of Population, Intervention, Comparator and Outcome method. (S1 PRISMA Checklist)

The protocol was registered with the International Prospective Register of Systematic Reviews (PROSPERO CRD42021218674) and has been detailed previously [14].

The Rayyan application was used to retrieve, screen for duplicates and reference all articles [17].

## Interventions

Studies that investigated the implementation and or efficacy of any mHealth modality in managing diabetes mellitus were included. The mHealth interventions reviewed included individual behavior change, chronic disease self-management, clinic appointment reminders, or clinical diagnostic aids. mHealth strategies under scrutiny included but were not limited to SMS text messages, phone calls, customized mobile phone applications, and other components of mHealth.

## Outcome measures

The primary focus was assessing the effectiveness of incorporating mHealth interventions into treatment regimens and comparing HbA1c or FPG level changes between intervention and control groups. Secondary outcomes aimed to measure reductions in diabetes mellitus complications, but most studies did not address this, nor did they specifically investigate mHealth strategies targeting these outcomes.

## Study records

**Data management/selection process.** Reviewers screened titles and abstracts retrieved using the search strategy and from additional sources in multiples of two to select studies that met the inclusion criteria outlined above [FD and BU, EM and CE, JC and II, IS and HI].

The selected full-text articles were retrieved and independently reassessed for eligibility by the two authors (S2 Table). A third reviewer was brought in from another group to resolve any disagreement between the two authors in any group by discussion. The selected studies were then included for data extraction and analysis.

## Data collection process

A standardized Microsoft Excel form was used for data extraction, quality assessment of studies, and evidence synthesis. Data extracted from the articles selected included the names of authors, year of publication, country of research, sample sizes of both study and control groups, drug treatments used in each study, particular mHealth intervention used in the study with the strategy of delivery and functionalities involved, type and duration of studies, study methodology, outcomes and times of measurement including mean baseline glycemic control using HbA1c or fasting plasma glucose, the mean glycemic control at the end of the study using HbA1c and fasting plasma glucose, the percentage change in HbA1c, number of hypoglycemic events and hyperglycemic crises, quality of life and body mass index (BMI) changes; harms including loss of privacy and inappropriate clinical events; cost of implementation; and indicators of acceptability of intervention information for assessment of risk of bias.

## Risk of bias in selected studies

The methodological quality of the included articles was assessed independently initially by one author [FD] using the Revised Cochrane Risk of Bias tool for RCTs. Two other authors [EM] and [JCM] then vetted this. Disagreements were then resolved by discussion [18].

The assessments were conducted using standardized processes to ensure objectivity across various domains. These included evaluating the concealment of treatment allocation, blinding of personnel involved in outcome assessment and data analysis, completeness of outcome data, presence of selective outcome reporting, and identification of other potential sources of bias. The automatically generated conclusions from the Revised Cochrane risk of bias tool were used, increasing inter-assessor agreement. (S2 Data)

## Data synthesis

A narrative synthesis was used to describe the interventions used in the studies, including individual behavioral change, chronic disease self-management, clinic appointment reminders or clinical diagnostic aid. Similarly, included were the mHealth category used and the mobile phone functionalities employed, the characteristics of the sample population and the expected outcomes, especially changes in HbA1c, number of hypoglycemic events and hyperglycemic crises, quality of life and BMI changes. We expected to use risk ratios for dichotomous outcomes and standardized mean differences for continuous outcomes to summarize the intervention effects with extracted data. However, the paucity of eligible studies impeded these calculations. We extracted the main outcomes of interest where available, analyzed them, and presented them mainly as narratives alongside other constructed variables of interest.

Additionally, the results were expressed in a table (S1 Data) comparing outcomes across the included studies, summarizing critical conclusions in the evidence table. Meta-analysis was performed on the mean difference in HbA1c (in %) calculated from the difference in standard deviation of HbA1c before and after intervention in the eligible studies (S1 Fig). It was done using metan, the main community contributed Stata 18 meta-analysis command which uses standard methods, and findings summarized as forest plots.[19,20] The interpretation of heterogeneity metrics is as follows: $I^2$ of at least 75% - high heterogeneity, $I^2$ between 50-75% - moderate heterogeneity, and $I^2$ of less than 50% - little to no heterogeneity [21].

## Results

The flowchart of the search results is shown in Fig 1. The search strategy yielded 1483 articles in all [MEDLINE = 304, EMBASE = 591, Web of Science = 243, Cochrane review = 150, Global Health citations = 102, CINAHL 91], of which only six were included. A total of 443

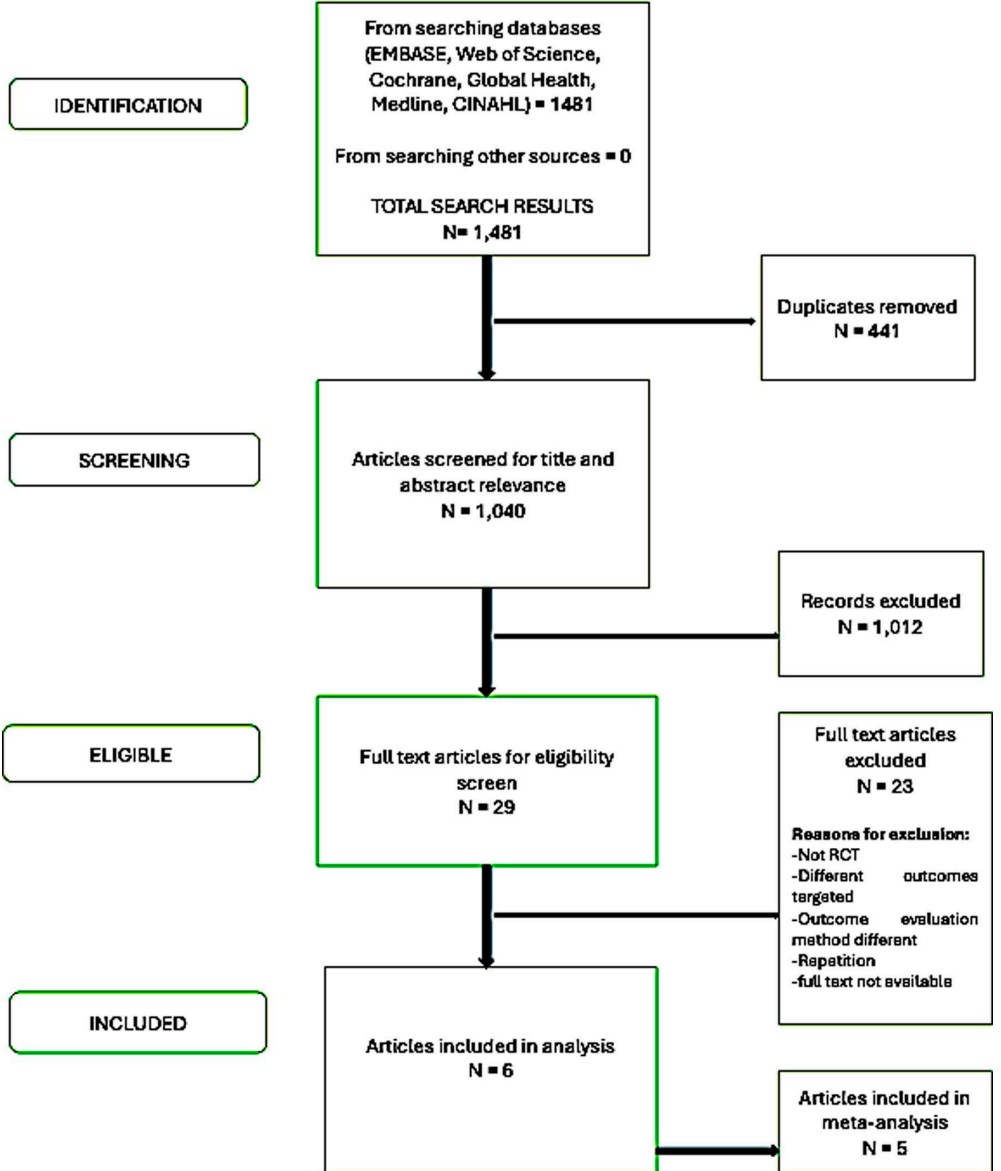

**Fig 1. The flowchart of the search results is shown.** The search strategy yielded 1481 articles in all. A total of 443 duplicates were removed. 29 articles screened for full-text eligibility after title and abstract screening. Assessment of the full-text articles resulted in the exclusion of an additional 23 articles, leaving a final total of 6 studies that met the eligibility criteria for the systematic review.

duplicates were removed. After title and abstract screening, studies 1,012 titles were excluded, leaving 29 articles screened for full-text eligibility. Assessment of the full-text articles resulted in the exclusion of an additional 23 articles, leaving a final total of 6 studies that met the eligibility criteria for the systematic review.

## Characteristics of included studies

All six eligible studies were experimental in design and published between 2010 and 2020 and only examined patients with type 2 diabetes. Four eligible studies investigated mHealth

intervention using a similar mHealth delivery strategy (SMS text messaging) [22–25]. In contrast, Asante et al. used phone calls, and Takenga et al. used a mobile application [26,27]. All targeted the same intervention, modifying the patient's behavior and attitude to treatment. The eligible studies were conducted in Ghana, Egypt, South Africa, Senegal and the Democratic Republic of Congo. The characteristics of the eligible studies are presented in Table 1.

## Methodological quality of included studies

This was assessed using the revised Cochrane Risk of Bias tool.[28] All the included studies were randomized controlled trials with varying methodological rigor. Table 2 summarizes the Risk of Bias assessment for each study. The outcome under examination is the change in HbA1c over the trial duration.

## Efficacy of mhealth interventions and delivery strategies

Five of the six studies targeted behavioral change by reaching remotely using SMS text messages and phone calls [22–26], while one study, Takenga et al [27], combined this with a direct feedback system. One of the studies did not provide the desired primary outcome of either HbA1c or FPG but used random blood glucose as its primary outcome [24]. All the studies were hospital-based and interventional, with follow-up periods ranging from 8 weeks to 52 weeks. They studied only

**Table 1. Characteristics of the eligible studies.**

| Author | Year | Country | Study Aim | Study design | Sampling strategy | Sample size |
|---|---|---|---|---|---|---|
| Haitham et al.[22] | 2017 | Egypt | To examine the feasibility of SMS education among diabetic patients in Egypt and assess the impact of educational text messages, compared to traditional paper-based methods, on glycemic control and self-management behaviors. | Randomized controlled trial | None given | 90 |
| Asante et al. [26] | 2020 | Ghana | To evaluate the feasibility and effectiveness of a nurse-led mobile phone call intervention on glycemic management and adherence to self-management practices among patients with type 2 diabetes mellitus in Ghana. | Randomized controlled trial | None given | 60 |
| Wargny et al [23] | 2018 | Senegal | To evaluate the impact of mHealth on diabetes control. | Randomized controlled trial | Consecutive | 180 |
| Owolabi et al.[24] | 2020 | South Africa | To determine the efficacy, acceptability and feasibility of text messaging in improving glycaemic control and other clinical outcomes among individuals living with diabetes in low-resource settings in Eastern Cape, South Africa. | Randomized controlled trial | None given | 216 |
| Farmer et al. [25] | 2021 | South Africa | To test the effectiveness of SMS-text messaging in improving outcomes in adults with type 2 diabetes. | Randomized controlled trial | Sequential | 1186 |
| Takenga et al. [27] | 2014 | Democratic Republic of Congo | To involve patients in the treatment process and motivate them for active participation. | Randomized controlled trial | Sequential | 40 |

**Table 2. Risk of Bias assessment.**

| Author/year | Randomisation Process | Deviation from intended Intervention | Missing outcome Data | Measurement of the outcome | Selection of the reported result | Overall |
|---|---|---|---|---|---|---|
| Haitham et al. 2017[22] | Low | Low | low | Low | low | low |
| Asante et al 2020[26] | High | Low | low | Low | high | high |
| Wargny et al. 2018[23] | Low | Low | low | Low | high | low |
| Owolabi et al. 2020[24] | Low | Low | low | Low | low | low |
| Farmer et al. 2021[25] | Some concerns | Low | low | Low | low | low |
| Takenga et al. 2014[27] | Low | Low | low | Low | low | low |

type 2 diabetes mellitus population and were conducted between 2014 and 2020. With precise randomization techniques, a clearly defined sampling strategy was available in only three studies (Table 1) [23,25,27]. The study with the largest sample size of 1186 had a 91% response rate [25]. Drug treatments for enrolled patients were heterogeneous, including oral antihyperglycemic agents and insulin therapy. None of the included studies measured primary outcomes in mean FPG or percentage changes. Five studies measured the percentage change in HbA1c from baseline to the end of the study, while Owolabi et al measured the mean change in blood glucose levels [24]. The percentage change in HbA1c from the baseline ranged from 3.6% to 20.53%, achieving significance in the studies by Takenga et al., Wargny et al., and Asante et al [23,26,27]. A meta-analysis of the pooled sample of 5 studies with a total participant size of 3112 excluding the article by Owolabi et al. which measured random glucose as its primary outcome was done.

The meta-analysis results are summarized in the forest plot in Fig 2. In the meta-analysis the overall WMD (95% CI) was 0.992 (0.48, 1.50). This, in combination with a high z score of 3.822, p <0.001 suggests a statistically significant overall effect that is not likely due to chance. However, a considerable heterogeneity ($I^2$ = 63.9%, p = 0.026) was present implying that the observed effect may not be generalizable to all the studies due to differences in study characteristics in this case most likely sample size and duration of study.

None of the studies addressed the secondary outcomes of measuring the direct relationships between these mHealth interventions and the prevention and early detection of diabetes complications. All six studies showed indications for the acceptability of mHealth interventions through administered questionnaires. No details of harm were available for assessment, though Wargny et al [23]. stated the limitation imposed by the high cost of implementing its preferred mHealth strategy (Table 3). Most studies paid limited attention to the secondary outcomes related to complication prevention through early detection using these mHealth interventions and delivery strategies.

## Discussion

This study examined the implementation of mHealth solutions and their impact on diabetes management in Africa. The objective was to assess the degree of mHealth integration into diabetes care and to evaluate the available evidence regarding its effectiveness in this context. Notably, all eligible studies primarily focused on type 2 diabetes. The overall effect size in the

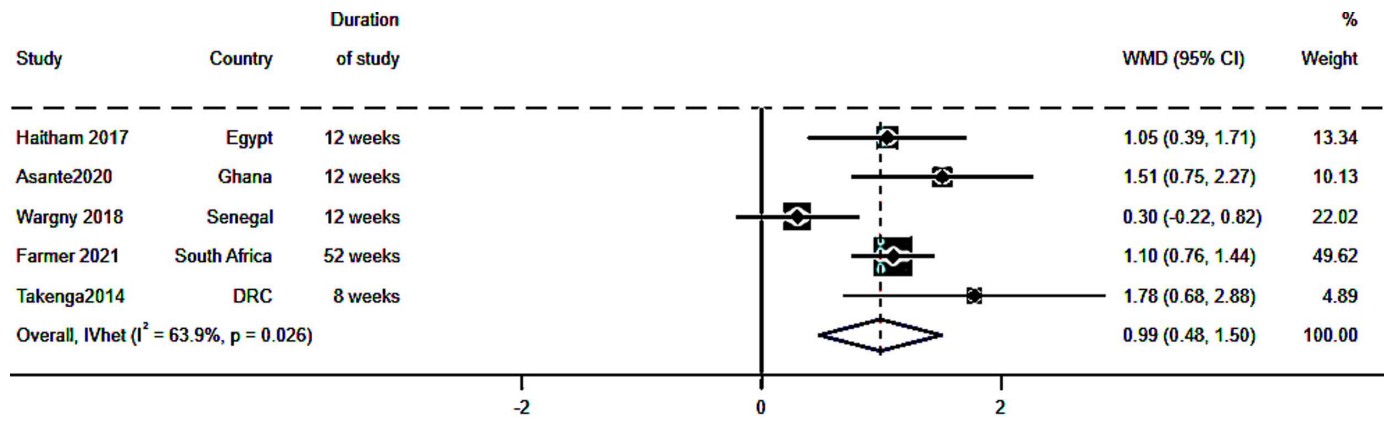

**Fig 2. The meta-analysis of the pooled sample of 5 studies with a total participant size of 3112 and the results are summarized in a forest plot.**

**Table 3. Summary of studies on interventions and efficacy of the mHealth delivery.**

| Author | Study aim | Study design | Study Time (wks) | mHealth tool/ strategy | mHhealth acceptance/ use/ response rate | mHhealth inter-vention outcomes | Significance in individ-ual studies | WMD with 95% CI from meta-analysis |
|---|---|---|---|---|---|---|---|---|
| Haitham et al.[22] | To examine the feasibility of SMS education among diabetic patients in Egypt, and assess the impact of educational text messages, compared to traditional paper-based methods, on glycemic control and self-management behaviors. | Ran-domized controlled trial | 12 | Reaching remotely via SMS/ call | 74.4% | 10.7% HbA1c reduction | no | 1.05 (0.39, 1.71) |
| Asante et al. [26] | To evaluate the feasibility and effectiveness of a nurse-led mobile phone call intervention on glycaemic management and adherence to self-management practices among patients with type 2 diabetes mellitus in Ghana. | Ran-domized controlled trial | 12 | Reaching remotely via call | 100% | 15.8% HbA1c reduction | Yes | 1.51 (0.75, 2.27) |
| Wargny et al[23] | To evaluate the impact of mHealth on diabetes control. | Ran-domized controlled trial | 12 | Reaching remotely via SMS/ call | 98% | 3.6% HbA1c reduction | Yes | 0.30 (-0.22, 0.82) |
| Owolabi et al.[24] | To determine the efficacy, acceptability and feasibility of text-messaging in improving glycaemic control and other clinical outcomes among individuals living with dia-betes in low-resource settings in Eastern Cape, South Africa. | Ran-domized controlled trial | 24 | Reaching remotely via SMS/ call | 100% | | NA | |
| Farmer et al.[25] | To test the effectiveness of SMS-text messaging in improving outcomes in adults with type 2 diabetes. | Ran-domized controlled trial | 52 | Reaching remotely via SMS/ call | 91% | | no | 1.10 (0.76, 1.44) |
| Takenga et al.[27] | To involve patients in the treatment process and motivate them for active participation | Ran-domized controlled trial | 8 | Direct feedback | 100% | 20.53% HbA1c reduction | Yes | 1.78 (0.68, 2.88) |

meta-analysis was statistically significant, suggesting mHealth may be beneficial in the man-agement of diabetes. In the meta-analysis of the included studies, almost half of the percentage weight is attributed to the study by Farmer et al., followed closely by Wargny et al. The relation-ship appears to be directly related to the sample size and the duration of the study. In contrast, the HbA1c percentage reduction declared by the individual studies is inversely associated with the sample size (Table 3). Takenga et al. reported a %HbA1c reduction of 20.53% with a sample size of 40, likely due to publication bias [27]. This study's findings serve as a stimulus for future research on mHealth implementation in diabetes mellitus management in Africa compared to other continents, where implementation has exceeded 87% in some high-income countries [1]. Assessing the efficacy and impact of mHealth becomes challenging without the implementation of these strategies. Therefore, more studies are needed in Africa to address this gap.

There is a lack of evidence on mHealth utilization, impact, and efficacy in Africa, rendering it impossible to compare these results with current evidence. However, the available global surveys of mHealth implementation by the WHO noted a higher strategic implementation of mHealth in high-income countries (HICs) [1]. It reached as high as 87% across all mHealth categories, starkly contrasting the notably low implementation levels observed in Africa during the same survey. Only one mHealth strategy was implemented, with a low phone penetration of 47% [1]. This paucity is confirmed in a global systematic review by Mao et al., which studied the global impact of mHealth in the management of diabetes mellitus and hypertension [29]. Out of 51 studies, including from 1747 articles, only 15 were conducted in developing countries, two from Africa. The review included the study by Haitham et al from Egypt which is included in our review, and another study on hypertension from South Africa.

They demonstrated that integrating mHealth into care positively impacted countries with different levels of economic development and that better outcomes follow the combination of mHealth with human intelligence [29] Another systematic review by Stevens et al. studying the effectiveness of digital health technologies for patients with diabetes mellitus included 25 RCTs in its final analysis with 3,360 patients [30]. None of the included studies was conducted in Africa. It showed an overall more significant improvement in HbA1c directly attributed to using a mHealth intervention compared with usual care for patients with type 1 diabetes mellitus, type 2 diabetes mellitus and prediabetes [30]. The focus of their research was on diabetes specific apps which came with the advent of smart phones. It is hoped that as smart phones become more popular in Africa such studies will also become available in keeping with the trend. The study by Bonoto et al found that the use of apps by diabetic patients showed significant effectiveness in reducing their hemoglobin A1c as well as strengthening their perception to self-care [9].

In our review, three of the six studies demonstrated a significant effect of mHealth on glycemic control by reducing HbA1c [23,26,27], with a statistically significant overall effect after meta-analysis. This finding is consistent with global findings reported by Mao et al. and Stevens et al which state clear benefit [29,30]. They are also consistent with findings of a recently published scoping review by Eberle et al., which reported that diabetes-specific mHealth applications significantly enhanced glycemic control, evidenced by reductions in HbA1c levels among individuals with type 1 and type 2 diabetes mellitus [11].

Despite its limitations, this review provides essential evidence to guide policies directed at increasing the adoption of mHealth interventions for managing diabetes mellitus in Africa, a continent grappling with poor healthcare systems and the high burden of communicable diseases.

## Implications for future research

While mobile phone adoption appears to be on the rise in Africa, [4,31] robust healthcare systems are lacking and mHealth interventions for managing diabetes mellitus remain underutilized, unlike in high-income countries, where various mHealth strategies are being deployed for diabetes self-management [1]. While the vast disparity in the uptake and implementation of mHealth interventions may be due to limited resources, awareness, and research, it is hoped that the outcome of this study will promote deliberate strategic policymaking by policymakers geared at enhancing the implementation of mHealth interventions in the delivery of diabetes mellitus care.

We recommend more studies on the effectiveness and benefits of carefully selected diabetes-specific mHealth apps through well-structured randomized controlled trials with well-defined targets comparable across correctly classified African diabetic populations. This would form a basis for more concrete conclusions from future systematic reviews examining the implementation and impact of this currently favored eHealth component in the continent.

## Study limitations and strengths

Only six studies were eligible for inclusion in this systematic review, indicating a limited evidence base and a clear gap in robust research concerning mHealth applications in the management of diabetes mellitus in Africa. Furthermore, it is noteworthy that many of the mHealth tools evaluated in these studies were outdated, particularly those relying on SMS text messaging. These tools have inherent limitations, such as the inability to deliver real-time data trends, susceptibility to data loss, and a lack of alert systems for patients experiencing hypoglycemia or severe hyperglycemia. In contrast, modern continuous glucose monitoring systems offer more advanced functionalities that could enhance patient management and outcomes [32,33]. The strategy of remotely reaching out through SMS messaging is prone to weak interaction

and technology challenges. With the greater adoption of smartphones worldwide and even in Africa, the interactive features of well-programmed phone applications are expected to be better harnessed to assist people living with diabetes mellitus in self-managing their condition. In this app/post-app era powered by smartphones, mHealth applications for specific diseases have become more valuable as operating systems advance and wearables and sensors are more widely used [5]. So, consumers now have the privilege of getting direct feedback from the apps. This is expected to revolutionize mHealth as a strategy for healthcare delivery, as predicted by Istepanian et al [2]. Studies testing the use of diabetes-specific mHealth applications are available, and have found these to be useful in promoting self-management in patients with diabetes [34,35]. Finally, as suggested above, the included studies may have been prone to publication bias.

## Conclusion

Overall, there was a statistically significant reduction in HbA1c levels among individuals living with type 2 diabetes in Africa following mHealth interventions. This is consistent with the findings of global systematic reviews. Although the effect may not be generalizable due to considerable heterogeneity, it remains a significant finding worthy of further exploration. We recommend conducting more well-designed randomized controlled trials to investigate this effect on both type 1 and type 2 diabetes mellitus. Additionally, the current findings can serve as a framework for policy development that integrates mHealth into diabetes management in Africa.

## Supporting information

**S1 Table. All studies.** Table of all the articles yielded by the search strategy described above it, prior to title and abstract screening. It shows all the yields from the databases searched.
(DOCX)

**S2 Table. Selected Full-text articles.** Table of selected full text articles after abstract and title screening.
(DOCX)

**S1 Data. Data table for numeric values extracted from study reports to calculate study effect estimates and measures of precision.**
(XLSX)

**S2 Data. Risk of bias.** Table of data for risk of bias summary using the Cochrane Risk of bias tool.
(XLSM)

**S1 Fig.** The complete meta-analysis tables, and the text summary of the derived indices.
(TIF)

## Author contributions

**Conceptualization:** Franklin Okechukwu Dike, Jean Claude Mutabazi, Ezekiel Musa, Blessing Chinenye Ubani, Ahmed Sherif Isa, Chidiebele Malachy Ezeude, Henry Iheonye, Isah Idris Ainavi.

**Data curation:** Franklin Okechukwu Dike, Jean Claude Mutabazi, Ezekiel Musa, Blessing Chinenye Ubani, Ahmed Sherif Isa, Chidiebele Malachy Ezeude, Henry Iheonye, Isah Idris Ainavi.

**Formal analysis:** Franklin Okechukwu Dike, Jean Claude Mutabazi, Ezekiel Musa, Blessing Chinenye Ubani, Ahmed Sherif Isa.

**Methodology:** Franklin Okechukwu Dike, Jean Claude Mutabazi, Ezekiel Musa, Blessing Chinenye Ubani, Henry Iheonye, Isah Idris Ainavi.

**Project administration:** Jean Claude Mutabazi, Ezekiel Musa.

**Resources:** Jean Claude Mutabazi, Ezekiel Musa.

**Software:** Jean Claude Mutabazi.

**Supervision:** Jean Claude Mutabazi, Ezekiel Musa.

**Visualization:** Franklin Okechukwu Dike.

**Writing – original draft:** Franklin Okechukwu Dike.

**Writing – review & editing:** Franklin Okechukwu Dike, Jean Claude Mutabazi, Ezekiel Musa, Blessing Chinenye Ubani, Ahmed Sherif Isa, Chidiebele Malachy Ezeude, Henry Iheonye, Isah Idris Ainavi.

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
