## [Decision Letter · Decision Letter 0]

11 Dec 2023

PDIG-D-23-00388

IMPLEMENTATION AND IMPACT OF mHEALTH IN THE MANAGEMENT OF DIABETES MELLITUS IN AFRICA: A SYSTEMATIC REVIEW

PLOS Digital Health

Dear Dr. Dike,

Thank you for submitting your manuscript to PLOS Digital Health. After careful consideration, we feel that it has merit but does not fully meet PLOS Digital Health's publication criteria as it currently stands. Therefore, we invite you to submit a revised version of the manuscript that addresses the points raised during the review process.

Please submit your revised manuscript within 60 days Feb 09 2024 11:59PM. If you will need more time than this to complete your revisions, please reply to this message or contact the journal office at digitalhealth@plos.org. Please include the following items when submitting your revised manuscript:

We look forward to receiving your revised manuscript.

Kind regards,

Haleh Ayatollahi

Section Editor

PLOS Digital Health

Journal Requirements:

1. Please provide separate figure files in .tif or .eps format only and remove any figures embedded in your manuscript file. Please also ensure that all files are under our size limit of 10MB.

2. Tables should not be uploaded as individual files. Please remove these files and include the Tables in your manuscript file as editable, cell-based objects. For more information about how to format tables, see our guidelines:

https://journals.plos.org/digitalhealth/s/tables

Additional Editor Comments (if provided):

Reviewers' comments:

Reviewer's Responses to Questions

**Comments to the Author**

1. Does this manuscript meet PLOS Digital Health’s publication criteria ? Is the manuscript technically sound, and do the data support the conclusions? The manuscript must describe methodologically and ethically rigorous research with conclusions that are appropriately drawn based on the data presented.

Reviewer #1: Partly

Reviewer #2: Yes

Reviewer #3: Partly

Reviewer #4: Yes

2. Has the statistical analysis been performed appropriately and rigorously?

Reviewer #1: N/A

Reviewer #2: Yes

Reviewer #3: Yes

Reviewer #4: Yes

3. Have the authors made all data underlying the findings in their manuscript fully available (please refer to the Data Availability Statement at the start of the manuscript PDF file)?

Reviewer #1: No

Reviewer #2: Yes

Reviewer #3: Yes

Reviewer #4: Yes

4. Is the manuscript presented in an intelligible fashion and written in standard English?

PLOS Digital Health does not copyedit accepted manuscripts, so the language in submitted articles must be clear, correct, and unambiguous. Any typographical or grammatical errors should be corrected at revision, so please note any specific errors here.

Reviewer #1: Yes

Reviewer #2: Yes

Reviewer #3: Yes

Reviewer #4: Yes

5. Review Comments to the Author

Please use the space provided to explain your answers to the questions above. You may also include additional comments for the author, including concerns about dual publication, research ethics, or publication ethics. (Please upload your review as an attachment if it exceeds 20,000 characters)

Reviewer #1: The paper by Dike et al. addresses an important issue, the feasibility and effectiveness of mobile health services for diabetics in Africa. This systematic review was originally planned to be conducted as a meta-analysis. However, no analysis was conducted due to the low number of studies found.

The authors are correct in recommending the need for more trials, but they fail to address the major criteria for publication, as well as some other important considerations.

Firstly, the authors did not seem to follow the journal’s guidelines as described here: https://journals.plos.org/digitalhealth/s/submission-guidelines#loc-systematic-reviews-and-meta-analyses

Notably, there is no numbering of the lines, references have not been correctly cited, and there is no author's summary. There is also no mention of the countries of affiliations. A major problem is that the main articles mentioned in the manuscript, the six that were retrieved, are not included in the references section, making it difficult to review the paper. I also believe that the authors should mention that they only refer to type 2 diabetes in the title and throughout the manuscript.

Other notes per sections in the manuscript:

Introduction

While the introduction is comprehensive and nicely summarizes the literature and the objectives of the paper, it is a bit unclear the way it is structured (there is no separation of paragraphs), and perhaps longer than needed. I feel that from “mHealth employs…” until “...GOe survey in 2005” is a bit too much of information and should be reduced to make it clearer.

Methods

The methods section is very clear and gives a good idea of the search strategy the authors used. Minor suggestions include moving the “Intervention” and “Outcomes” sections after the “Search strategy and selection criteria” section. In addition, the authors should add a reference after mentioning the WHO and IDF guidelines in the first section.

Results

In my opinion, this section needs to be substantially revised. As mentioned earlier, the primary articles of the review are not referenced, which makes understanding its scope difficult. It is also important to note that, since only two studies reported significant outcomes, table 3 may be misleading as it indicates the average outcome of the intervention. The average should at the very least be noted along with the confidence intervals so that readers can determine the nature of the results. Additionally, these non-significant studies should be mentioned in the text with caution. The duration of each intervention in the table should also be noted since there are only six studies. A simple description of their length of time between 8 and 52 weeks is insufficient to provide a comprehensive view of them.

Other minor comments: 

1. There are no references to the supplementary files in the text.

2. In the first paragraph, first mention how many articles were found (1,443) and then mention how many were excluded (1,012) and how many left.

3. In the text before table 3 “One of the studies did not…” – need to mention which one it is.

4. Also mention which studies found significance and which did not.

Discussion

While the implications in this section are compelling, it would be useful to elaborate on why more of these studies should be conducted in Africa. Additionally, this section lacks references. In addition, I believe that since most people intend to use smartphones in 2023 and thus applications for diabetes, it would be worthwhile to mention the limitations of the SMS/call approach in the discussion, since smartphone applications may prove to be more useful in treating T2D.

Conclusion

In my opinion, this review does not provide strong evidence that mHealth is beneficial in improving HbA1c for diabetics in Africa (only six studies were reviewed, and only two of those showed significance). According to this study, the only main conclusion is that more RCTs are necessary, primarily because there is a lack of evidence at this time.

Reviewer #2: Article is well written, statistically sound and supporting information is accessible though macros in Excel are blocked by microsoft on my machine. It is unfortunate but illuminating that there is a dearth of published studies in this field. Refer to attached PDF for minor proposals

Reviewer #3: The authors present the results of a rigorous systematic literature review. For the huge amount of work that went into this systematic literature review, I was surprised about the shortness of the results. Would it be possible to add some graphs or report some other characteristics of the studies, for example some baseline characteristics of the study populations? Although the authors write that there were not enough data, I was still wondering if it would be possible to perform a meta-regression on the outcome? The outcome of two references is missing in Table 3, so I assume it was not HbA1c but another endpoint. By using the ratio of means instead of the mean the difference in outcomes could be overcome. Then there would be 6 studies included in the meta-analysis, which seems to be a reasonable number despite high heterogeneity. I also miss a visualisation in this manuscript. Please consider adding one. For example, a plot including the ratio of means (y-axis) in dependence with the duration of follow-up (x-axis), maybe with different sizes of the markers to visualise the different sample sizes, would be interesting. Or a boxplot visualising differences in interventional and control groups. In addition, or alternatively, would it be feasible to compare the African context with another one, e.g. Latinamerican, Asian, and then perform a stratified meta-analysis? 

These are only a few suggestions that might increase the impact and quality of the manuscript. 

Minor comments: 

Affiliations: please add country to the affiliations

Introduction: 

I suggest writing clearly that this article is about type 2 diabetes and does not refer to type 1 diabetes. 

It is not clear if the sentences following “It showed that mHealth was only just emerging in many member states...” are referring to the present situation or the survey in 2005; please clarify in the text. 

I suggest writing the objectives in a dedicated paragraph instead of listing them up. 

Methods:

Outcome measures: please introduce the abbreviation FPG before the first use

Outcome measures: what is the conventional therapy?

Outcome measures: “However, most of the studies included...” This is a result and should not be in the methods section. 

Data Synthesis: what ist meant by “expressed statistically”? Please rephrase

Results: 

Characteristics of included studies: “None of the observational studies met the eligibility criteria.” Why is this sentence here, as the paragraph refers to the characteristics of the included studies?

Characteristics of included studies: “All the eligible studies investigated a mHealth intervention using a similar mHealth delivery strategy [SMS text messaging]…” “all” meaning four out of six?

Table 1: it would be nice to see the sample size. In addition, it would be nice to add the information regarding the mHealth delivery strategy in a separate column for better overview. 

Efficacy of the interventions and efficacy of the mHealth delivery strategies: “Five of the six studies…” which study are you referring to? Please add the reference. The same applies to the next sentence. Idem for “The study with the largest sample size of 1186 had a 91% response rate.”

Table 1 and Table 3 contain almost the same information, please get rid of the redundant information. 

Table 3: what was the outcome in Owolabi and what in Farmer? 

Conclusion: 

The study does not support the conclusion “The study shows mHealth is beneficial in improving glycemic control (HbA1c) in patients with diabetes mellitus and its utilization could be essential in preventing and or delaying diabetes complications.” We would need to see the results of a meta-analysis to assess this.

Reviewer #4: Thank you for this important Review concerning mHealth in Diabetes in Africa. Only six randomized trials was selected in this carefully performed process. This is really well done and the paper is ready to get published. Hopefully we will get more data within this field in the future, mHealth is probably a most important tool.

6. PLOS authors have the option to publish the peer review history of their article (what does this mean? ). If published, this will include your full peer review and any attached files.

**Do you want your identity to be public for this peer review?** For information about this choice, including consent withdrawal, please see our Privacy Policy .

Reviewer #1: Yes: Shlomo Yeshurun

Reviewer #2: Yes: Eric Mugambi

Reviewer #3: No

Reviewer #4: No

---

## [Decision Letter · Decision Letter 1]

7 May 2024

PDIG-D-23-00388R1

IMPLEMENTATION AND IMPACT OF mHEALTH IN THE MANAGEMENT OF DIABETES MELLITUS IN AFRICA: A SYSTEMATIC REVIEW AND META-ANALYSIS

PLOS Digital Health

Dear Dr. Dike,

Thank you for submitting your manuscript to PLOS Digital Health. After careful consideration, we feel that it has merit but does not fully meet PLOS Digital Health's publication criteria as it currently stands. Therefore, we invite you to submit a revised version of the manuscript that addresses the points raised during the review process.

Please submit your revised manuscript within 60 days Jul 06 2024 11:59PM. If you will need more time than this to complete your revisions, please reply to this message or contact the journal office at digitalhealth@plos.org. Please include the following items when submitting your revised manuscript:

We look forward to receiving your revised manuscript.

Kind regards,

Haleh Ayatollahi

Section Editor

PLOS Digital Health

Journal Requirements:

Additional Editor Comments (if provided):

Reviewers' comments:

Reviewer's Responses to Questions

**Comments to the Author**

1. If the authors have adequately addressed your comments raised in a previous round of review and you feel that this manuscript is now acceptable for publication, you may indicate that here to bypass the “Comments to the Author” section, enter your conflict of interest statement in the “Confidential to Editor” section, and submit your "Accept" recommendation.

Reviewer #1: (No Response)

Reviewer #2: (No Response)

Reviewer #3: (No Response)

2. Does this manuscript meet PLOS Digital Health’s publication criteria ? Is the manuscript technically sound, and do the data support the conclusions? The manuscript must describe methodologically and ethically rigorous research with conclusions that are appropriately drawn based on the data presented.

Reviewer #1: Partly

Reviewer #2: Yes

Reviewer #3: Yes

3. Has the statistical analysis been performed appropriately and rigorously?

Reviewer #1: Yes

Reviewer #2: I don't know

Reviewer #3: No

4. Have the authors made all data underlying the findings in their manuscript fully available (please refer to the Data Availability Statement at the start of the manuscript PDF file)?

Reviewer #1: Yes

Reviewer #2: No

Reviewer #3: Yes

5. Is the manuscript presented in an intelligible fashion and written in standard English?

PLOS Digital Health does not copyedit accepted manuscripts, so the language in submitted articles must be clear, correct, and unambiguous. Any typographical or grammatical errors should be corrected at revision, so please note any specific errors here.

Reviewer #1: Yes

Reviewer #2: Yes

Reviewer #3: Yes

6. Review Comments to the Author

Please use the space provided to explain your answers to the questions above. You may also include additional comments for the author, including concerns about dual publication, research ethics, or publication ethics. (Please upload your review as an attachment if it exceeds 20,000 characters)

Reviewer #1: Thank you for submitting the revised manuscript and I appreciate the efforts on my previous comments. While there have been some improvements, there are still several critical issues that need to be addressed before the manuscript can be considered for publication. Please address these issues thoroughly to enhance the manuscript's clarity and accuracy.

Major

While I understand that the aim of the study was to search for both types of diabetes, since the 6 studies only tested type 2 diabetes, all results, discussion, and conclusions section should highlight this fact.

Furthermore, there are still many unsuitable references throughout the manuscript. Notably, in line 134 (introduction) the references do not support the statement. I believe the authors should look carefully at each reference and its relevance, in particular in the introduction and discussion sections.

Please mention if achieved significant for Hba1c for each paper mentioned in the results section under “EFFICACY OF THE INTERVENTIONS AND EFFICACY OF THE MHEALTH DELIVERY STRATEGIES”. I believe this is important to understand these studies with cautious.

I believe that after a comprehensive analysis about this important topic, the discussion should be expanded, mostly with suggestions about future studies and how they should be conducted with relevance to this day and age. In addition, the strength and limitation’s part should be much further explained. In particular they should mention the large differences between the six studies in their population, length, methodology, and results. They could also mention the limitations of SMS and why smartphone apps can do a better job.

While the authors responded that the conclusion part has been edited, I don’t see any changes. I think that since only 6 studies have been reviewed with large differences between them, words like “demonstrates” and “essential” should be avoided. In my understanding, the main conclusion from this review is that future research is urgently needed since it’s impossible to conclude anything about the impact of mHealth for diabetes in Africa at the current stage.

Minor

The Author’s Summary is well-written and a nice addition to the manuscript. Still, I would suggest looking at similar articles in PLOS Digital Health as it should be more concise and in one paragraph.

While the authors made some changes to the introduction, they have not addressed my comment regarding lines 116 to 120. Please consider modifying this. 

In addition, I believe that most cited papers are outdates regarding the study aims. Since the manuscript it about Digital Health, it’s important to look for more recent surveys and meta-analysis as technology advanced rapidly since (most cited studies are before the age of smartphones).

In the results section, please add a reference each time a study is mentioned (line 315 for example).

Thank you for adding the extra information in the tables. Please consider merging tables 1 and 3 into one table as some of the information is repeatable. Also, the heading of the first column in table 3 does not reflect the contents. I would also suggest another column of “Significance” with Yes/No for it.

I still don’t see a reference in the manuscript for the supplementary files. Please make sure that they are mentioned correctly.

Reviewer #2: I am unable to access the data underlying the findings as a result of a technical glitch on PDF that keeps downloading a copy of PRISMA checklist for all the different items clicked.

The manuscript is coherent but does not make for easy reading - I have provided examples of how certain critical area can be rephrased. It may be useful to have a technical writer improve the flow of text without altering the core structure or message.

Reviewer #3: Comments regarding meta-analysis: 

Thank you for doing a meta-analysis. It would be nice if the authors would describe the meta-analysis in more detail in the “data synthesis” section. On what was the meta-analysis done (mean difference of what); using which statistical package was the meta-analysis done. The authors should explain how metrics of heterogeneity will be interpreted (e.g. I^2). I suggest adding the corresponding reference (e.g. Higgins JPT, Thompson SG, Deeks JJ, Altman DG. Measuring inconsistency in meta-analyses. BMJ 2003; 327: 557–60.)

In the forest plot of the meta-analysis there is a missing space between study author and year of publication for Asante and Takenga. The abbreviation DRC and WMD should be explained in the figure legend. I suggest to also add the sample size in the graph. 

In the flowchart, it says “articles included in quantitative analysis = 6”. What is meant by “quantitative analysis” and how does this differ from meta-analysis? The flowchart should also include the meta-analysis (by adding one more row for example). 

Other comments: 

The abbreviation “DM” is used without introducing it (in the main text; it seems that the introduction has been deleted in the revised version of the manuscript)

It is still not clear that the systematic review and meta-analysis are referring to type 2 diabetes. I suggest adding “type 2” in the title

Lines 356-358 is difficult to understand; please review.

7. PLOS authors have the option to publish the peer review history of their article (what does this mean? ). If published, this will include your full peer review and any attached files.

**Do you want your identity to be public for this peer review?** For information about this choice, including consent withdrawal, please see our Privacy Policy . 

Reviewer #1: Yes: Shlomo Yeshurun

Reviewer #2: Yes: Eric Mugambi Nturibi

Reviewer #3: No

---

## [Decision Letter · Decision Letter 2]

9 Aug 2024

PDIG-D-23-00388R2

IMPLEMENTATION AND IMPACT OF mHEALTH IN THE MANAGEMENT OF DIABETES MELLITUS IN AFRICA: A SYSTEMATIC REVIEW AND META-ANALYSIS

PLOS Digital Health

Dear Dr. Dike,

Thank you for submitting your manuscript to PLOS Digital Health. After careful consideration, we feel that it has merit but does not fully meet PLOS Digital Health's publication criteria as it currently stands. Therefore, we invite you to submit a revised version of the manuscript that addresses the points raised during the review process.

Please submit your revised manuscript within 60 days Oct 08 2024 11:59PM. If you will need more time than this to complete your revisions, please reply to this message or contact the journal office at digitalhealth@plos.org. Please include the following items when submitting your revised manuscript:

We look forward to receiving your revised manuscript.

Kind regards,

Haleh Ayatollahi

Section Editor

PLOS Digital Health

Journal Requirements:

Additional Editor Comments (if provided):

Reviewers' comments:

Reviewer's Responses to Questions

**Comments to the Author**

1. If the authors have adequately addressed your comments raised in a previous round of review and you feel that this manuscript is now acceptable for publication, you may indicate that here to bypass the “Comments to the Author” section, enter your conflict of interest statement in the “Confidential to Editor” section, and submit your "Accept" recommendation.

Reviewer #1: (No Response)

Reviewer #2: All comments have been addressed

Reviewer #3: (No Response)

2. Does this manuscript meet PLOS Digital Health’s publication criteria ? Is the manuscript technically sound, and do the data support the conclusions? The manuscript must describe methodologically and ethically rigorous research with conclusions that are appropriately drawn based on the data presented.

Reviewer #1: Partly

Reviewer #2: Yes

Reviewer #3: Partly

3. Has the statistical analysis been performed appropriately and rigorously?

Reviewer #1: Yes

Reviewer #2: Yes

Reviewer #3: No

4. Have the authors made all data underlying the findings in their manuscript fully available (please refer to the Data Availability Statement at the start of the manuscript PDF file)?

Reviewer #1: Yes

Reviewer #2: Yes

Reviewer #3: Yes

5. Is the manuscript presented in an intelligible fashion and written in standard English?

PLOS Digital Health does not copyedit accepted manuscripts, so the language in submitted articles must be clear, correct, and unambiguous. Any typographical or grammatical errors should be corrected at revision, so please note any specific errors here.

Reviewer #1: Yes

Reviewer #2: Yes

Reviewer #3: No

6. Review Comments to the Author

Please use the space provided to explain your answers to the questions above. You may also include additional comments for the author, including concerns about dual publication, research ethics, or publication ethics. (Please upload your review as an attachment if it exceeds 20,000 characters)

Reviewer #1: Dear Authors,

I appreciate the efforts you have made in revising the manuscript. However, there are several points from my previous comments that remain unaddressed. Below are my detailed comments:

Abstract

1) The use of the term “trend” should not be used. Please mention if reached significance or not. Please clearly indicate whether the meta-analysis results are statistically significant. This is crucial for understanding the validity of your findings.

2) The conclusions section:

Consider making it more concise to fit the abstract format.

Ensure that the conclusions in the abstract match exactly with those in the discussion section for consistency.

It is essential to explicitly mention type 2 diabetes in the conclusions to avoid any ambiguity regarding the scope of your study.

Authors’ Summary

This section still does not conform to the style and format typical of similar articles in PLOS Digital Health. It should be more concise and structured into a single paragraph. I urge you to revise this section to meet these standards.

Introduction

1) For the second time, the introduction has not been updated as requested. The first and second paragraphs contain outdated information and do not reflect the latest advancements and definitions in mHealth as of 2024. It is imperative to include recent surveys, definitions, and relevant manuscripts to provide a current context.

2) While the changes in the rest of the introduction are appreciated, the paragraph before the last (line 163) feels disconnected from the rest of the introduction. Please either expand on this subject to integrate it seamlessly with the other paragraphs or consider relocating it to the discussion section where it might be more relevant.

Results

3) There are still instances where studies are mentioned without proper references, particularly in lines 325 and 326. It is crucial to provide references for all mentioned studies to ensure the credibility and traceability of your work.

4) Thank you for adding the significance of the findings. However, it is still unclear whether the significance indicated in Table 3 pertains to the WMD or if it was determined within the individual studies. Please clarify this in the table for better understanding.

Discussion

1) While the discussion section has been expanded, it still requires further enhancement. Specifically, I recommend including detailed suggestions for future studies, highlighting how they should be conducted in the context of current technological advancements and healthcare needs.

2) The strengths and limitations section needs a more comprehensive explanation. It is important to discuss the significant differences among the six studies regarding their population, length, methodology, and results. Furthermore, addressing the limitations of SMS-based interventions and discussing why smartphone applications might offer better solutions would add significant value to this section.

I hope these detailed comments provide clear guidance for your revisions. Ensuring these points are adequately addressed will greatly enhance the quality and impact of your manuscript.

Many thanks

Reviewer #2: The Authors have satisfactorily adjusted the article based on my recommendations.

Reviewer #3: Unfortunately the authors didn't address my previous comments. Stata is not a statistical package, but a statistical software. The authors should add the reference (can be found via google) and explain which statistical package/function/command they used for their meta-analysis (can also be found via google). The interpretation of the I^2 should also include a reference (pubmed!)

Figure 2 is still not adequate (missing spaces between author and year). 

lines 324/380/450: I don't think "tested" is the correct wording in this context, as it is likely the app that was assessed. What about "included"?

7. PLOS authors have the option to publish the peer review history of their article (what does this mean? ). If published, this will include your full peer review and any attached files.

**Do you want your identity to be public for this peer review?** For information about this choice, including consent withdrawal, please see our Privacy Policy . 

Reviewer #1: Yes: Shlomo Yeshurun

Reviewer #2: No

Reviewer #3: No

---

## [Decision Letter · Decision Letter 3]

19 Nov 2024

PDIG-D-23-00388R3IMPLEMENTATION AND IMPACT OF mHEALTH IN THE MANAGEMENT OF DIABETES MELLITUS IN AFRICA: A SYSTEMATIC REVIEW AND META-ANALYSISPLOS Digital Health Dear Dr. Dike, Thank you for submitting your manuscript to PLOS Digital Health. After careful consideration, we feel that it has merit but does not fully meet PLOS Digital Health's publication criteria as it currently stands. Therefore, we invite you to submit a revised version of the manuscript that addresses the points raised during the review process. Please submit your revised manuscript within 30 days Dec 19 2024 11:59PM. If you will need more time than this to complete your revisions, please reply to this message or contact the journal office at digitalhealth@plos.org. Please include the following items when submitting your revised manuscript:* A rebuttal letter that responds to each point raised by the editor and reviewer(s). You should upload this letter as a separate file labeled 'Response to Reviewers '. This file does not need to include responses to any formatting updates and technical items listed in the 'Journal Requirements' section below.* A marked-up copy of your manuscript that highlights changes made to the original version. You should upload this as a separate file labeled 'Revised Manuscript with Track Changes '.* An unmarked version of your revised paper without tracked changes. You should upload this as a separate file labeled 'Manuscript '. If you would like to make changes to your financial disclosure, competing interests statement, or data availability statement, please make these updates within the submission form at the time of resubmission. Guidelines for resubmitting your figure files are available below the reviewer comments at the end of this letter. We look forward to receiving your revised manuscript. Kind regards, Haleh AyatollahiSection EditorPLOS Digital Health Haleh AyatollahiSection EditorPLOS Digital Health Leo Anthony CeliEditor-in-ChiefPLOS Digital Healthorcid.org/0000-0001-6712-6626  **Journal Requirements:**

1. As required by our policy on Data Availability, please ensure your manuscript or supplementary information includes the following: 

**Additional Editor Comments (if provided):****Reviewers' Comments:** Reviewer's Responses to Questions

**Comments to the Author**

1. If the authors have adequately addressed your comments raised in a previous round of review and you feel that this manuscript is now acceptable for publication, you may indicate that here to bypass the “Comments to the Author” section, enter your conflict of interest statement in the “Confidential to Editor” section, and submit your "Accept" recommendation.

Reviewer #2: (No Response)

Reviewer #3: (No Response)

2. Does this manuscript meet PLOS Digital Health’s publication criteria ? Is the manuscript technically sound, and do the data support the conclusions? The manuscript must describe methodologically and ethically rigorous research with conclusions that are appropriately drawn based on the data presented.

Reviewer #2: Yes

Reviewer #3: Partly

3. Has the statistical analysis been performed appropriately and rigorously?

Reviewer #2: Yes

Reviewer #3: No

4. Have the authors made all data underlying the findings in their manuscript fully available (please refer to the Data Availability Statement at the start of the manuscript PDF file)?

Reviewer #2: Yes

Reviewer #3: Yes

5. Is the manuscript presented in an intelligible fashion and written in standard English?

Reviewer #2: Yes

Reviewer #3: No

6. Review Comments to the Author

Reviewer #2: The manuscript is shaping up but there are a number of adjustments the authors can make to improve its quality

Please include the reference for Rayyan as it is relatively new:

Mourad Ouzzani, Hossam Hammady, Zbys Fedorowicz, and Ahmed Elmagarmid. Rayyan — a web and mobile app for systematic reviews. Systematic Reviews (2016) 5:210, DOI: 10.1186/s13643-016-0384-4.

Table 1 - adjust column sizes - for readability so that words are not truncated

Table 2 truncated - last column

Table 3 - similar comments to other tables, for this table even names of authors are cut-off. Please rectify to improve the reader's experience

Line 395 could simply read "Efficacy of mHealth Interventions and delivery strategies - and not repeat the word "efficacy"

Line 399 - after et al and reference, there's a letter "t" left hanging

DISCUSSION

Lines 446-448 - Looking closely at the six 446, ' studies included in the meta-analysis, almost half of the percentage weight is attributed to the study by Farmer et al., with 49.62%, followed closely by Owolabi et al...'

This is a bit confusing to me as Owolabi et al was excluded from the meta-analysis having used RBS and not FBS/HBA1c - as indicated in lines 414 to 417

Standardize brackets style - you have used rectangular brackets [] in line 452 [Table 3] and round (paratheses) in other sections

Also as per PLOS-DH formatting guidelines: the Vancouver in text citations should be in square brackets e.g. [1], [2-5]. See this guide for suggested manuscript formatting rules.

https://storage.googleapis.com/plos-published-prod/9cba/PLOS%20Manuscript%20Body%20Formatting%20Guidelines.pdf?X-Goog-Algorithm=GOOG4-RSA-SHA256&X-Goog-Credential=wombat-sa%40plos-prod.iam.gserviceaccount.com%2F20241112%2Fauto%2Fstorage%2Fgoog4_request&X-Goog-Date=20241112T175512Z&X-Goog-Expires=86400&X-Goog-SignedHeaders=host&X-Goog-Signature=10a23077063042dde83b9996a5b631513f6772b7947b8a9c78ec92d94895c8fe830fe254e4a7c340e794200c8a53c671d7c3a726ee31fd47d834c6e689dbb31cb6b15f4f40b78c58c777e3d9147c5595a95c5080ae54f747f132c973ae7516bc7572cc528744d07ce38a95b953acdde41d8d22683b4f39a0e8f6d0fab682f47c9733ca984e5922802d44c76542fa12793d2657eec4c35e0116e0ec775150b1350ad24cd50a3b4fc2a2ae40f0d3fed7b22986b709e24ff52006b9dcc7762894c3c73dc3a78c190175d3cd92307d805b365043b00c3ef04994e52cc83e4113eec1078a0ce721b5ec193916c001517d4540b89a3a5914aab6486bb4fc0dc6e7f280

or here: https://shorturl.at/9h6T8

or here: https://journals.plos.org/digitalhealth/s/submission-guidelines#loc-methods-software-databases-and-tools

Line 462 - more accurate to say - with current published evidence - in place of 'with previous evidence.'I would do away with the word 'nearly' and stick to 'rendering it impossible' to make comparisons

Lines 458-459 - Perhaps add detail to propose the bare minimum for such studies in terms of methodology, patient population and outcomes - these would be informed by other seminal studies as uncovered by your literature review - WHO-mERA has good suggestions for digital health solutions: https://www.who.int/publications/i/item/9789241550505

DISCUSSION

Mao et al and Stevens et al in their SR did not include any of the studies you identified (or if they did they used different names) - please speak to this in your discussion - they may not have searched African study databases for instance...

Line 503 - the word "massively" is probably not appropriate as it might imply every facility/unit in the first world is deploying mHealth - yet the Global systemic reviews did not identify that many studies.

Does Bonoto et al have any bearing on your results that may be of value in your discussion? - you have restricted its use to the introductory sections

In your conclusion you have affirmed that the only available studies featured type 2 DM. In your concluding sentence, you might want to come out strongly that more well designed studies investigating m-Health interventions and strategies in patients with type 1 and 2 diabetes are urgently needed.

REFERENCES

1. https://www.who.int/goe/publications/goe_mhealth_web.pdf - for reference 1 - link not active

3.Reference 3 which is a book requires more information including year, publisher and place of publication

19 - I do not think reference 19 is necessary - FAQ: Citing Stata software, documentation, and FAQs | Stata [Internet]. [cited 2024 Nov 2]. Available from: https://www.stata.com/support/faqs/resources/citing-software-

documentation-faqs/ - Is it? moreover the link is not working.

If the authors wanted to highlight the use of the metan command (assuming this is not standard practice) - there are references e.g. Harris RJ, Deeks JJ, Altman DG, Bradburn MJ, Harbord RM, Sterne JA. Metan: fixed-and random-effects meta-analysis. The Stata Journal. 2008 Apr;8(1):3-28. - that may suffice - or they could provide a working link.

30 - This reference is not correctly formatted as per Vancouver style guidelines:

NW 1615 L. St, Suite 800Washington, Inquiries D 20036USA202 419 666

4300 | M 857 8562 | F 419 4372 | M. Basic mobile phones more common than smartphones in sub-Saharan Africa [Internet]. Pew Research Center’s Global Attitudes Project. 2018 [cited 2020 Oct 5]. Available from: https://www.pewresearch.org/global/2018/10/09/majorities-in-sub-saharan-africa-own-mobile-phones-but-smartphone-adoption-is-modest/

32 - the accurate reference is:

Albrecht UV, von Jan U. mHealth Apps and Their Risks–Taking Stock. In Unifying the Applications and Foundations of Biomedical and Health Informatics 2016 (pp. 225-228). IOS Press.

FIGURES

Resolution appears fine - confirm with PACE tool (https://pacev2.apexcovantage.com/)

On fig 1 - please stick to consistent paragraph formatting in the text boxes e.g. on the identification row - text is left aligned, also the N value for other sources which is 0 - is on the same line as the rest of the text, where as it is on the next separate line elsewhere

Second rung - total search results - the N value is "hanging" - bring it down to next line for consistency - decide whether to centre all the N values or apply a similar style e.g. left alignment

On the last row - centre - "Articles included..." N=6 - you have centered formatting here which is not applied anywhere else!

There's no reason the text boxes are all of different sizes and not aligned! Also stick to straight arrows as they are easy on the eye.

The metaanalysis you referenced by Mao et al has a flowchart you can revisit - https://pmc.ncbi.nlm.nih.gov/articles/instance/7523197/bin/bmjdrc-2020-001225supp002.pdf

Reviewer #3: Line 65/405/666: “… with a p value of <0.001”

Line 404: I don’t understand this sentence. How was this tested? “The test of overall effect within the meta-analysis was 0, with a p value of 0 suggesting that a null effect was found across the included studies”

Line 406: what does a WMD of 1 mean?

Line 408: If it is non-significant that does not mean that the study was not powered. Please adapt the phrasing. And why “in contrast”?

Personally, I find the conclusion “more high-quality research is needed” a bit dull. Which aspects could have led to these inconclusive study results? It is a reporting bias? Or because of the study design? What would be the ideal study design?

7. PLOS authors have the option to publish the peer review history of their article (what does this mean? ). If published, this will include your full peer review and any attached files.

**Do you want your identity to be public for this peer review?** For information about this choice, including consent withdrawal, please see our Privacy Policy .

Reviewer #2: **Yes: ** Eric Mugambi Nturibi

Reviewer #3: No

---

## [Decision Letter · Decision Letter 4]

7 Feb 2025

IMPLEMENTATION AND IMPACT OF mHEALTH IN THE MANAGEMENT OF DIABETES MELLITUS IN AFRICA: A SYSTEMATIC REVIEW AND META-ANALYSIS

PDIG-D-23-00388R4

Dear Dr Dike,

We are pleased to inform you that your manuscript 'IMPLEMENTATION AND IMPACT OF mHEALTH IN THE MANAGEMENT OF DIABETES MELLITUS IN AFRICA: A SYSTEMATIC REVIEW AND META-ANALYSIS' has been provisionally accepted for publication in PLOS Digital Health.

Best regards,

Haleh Ayatollahi

Section Editor

PLOS Digital Health

**Additional Editor Comments (if provided):**

**Reviewer Comments (if any, and for reference):**

Reviewer's Responses to Questions

**Comments to the Author**

1. If the authors have adequately addressed your comments raised in a previous round of review and you feel that this manuscript is now acceptable for publication, you may indicate that here to bypass the “Comments to the Author” section, enter your conflict of interest statement in the “Confidential to Editor” section, and submit your "Accept" recommendation.

Reviewer #2: All comments have been addressed

Reviewer #4: All comments have been addressed

2. Does this manuscript meet PLOS Digital Health’s publication criteria ? Is the manuscript technically sound, and do the data support the conclusions? The manuscript must describe methodologically and ethically rigorous research with conclusions that are appropriately drawn based on the data presented.

Reviewer #2: Yes

Reviewer #4: Yes

3. Has the statistical analysis been performed appropriately and rigorously?

Reviewer #2: Yes

Reviewer #4: Yes

4. Have the authors made all data underlying the findings in their manuscript fully available (please refer to the Data Availability Statement at the start of the manuscript PDF file)?

Reviewer #2: Yes

Reviewer #4: Yes

5. Is the manuscript presented in an intelligible fashion and written in standard English?

Reviewer #2: Yes

Reviewer #4: Yes

6. Review Comments to the Author

Reviewer #2: (No Response)

Reviewer #4: No more comments

7. PLOS authors have the option to publish the peer review history of their article (what does this mean? ). If published, this will include your full peer review and any attached files.

**Do you want your identity to be public for this peer review?** For information about this choice, including consent withdrawal, please see our Privacy Policy .

Reviewer #2: **Yes: ** ERIC MUGAMBI NTURIBI

Reviewer #4: No
